

# Observing long-lived longwave contrail forcing

Aaron Sonabend-W[1], Scott Geraedts[1], Nita Goyal[1], Joe Yue-Hei Ng[1], Christopher Van Arsdale[1], and
Kevin McCloskey[1]

[1]Google Research, Mountain View, CA, USA

**Correspondence:** Kevin McCloskey (mccloskey@google.com)

**Abstract.** Contrail microphysical simulations and climate simulations have indicated that contrail cirrus cause a substantial
fraction of aviation's climate impact. While the approximations and parameter selections in these simulations have been well-
validated over the past two decades, the heat trapping of contrails has not been observed using satellite data beyond a few
hours. This is because contrails lose their linear shape after a few hours, making them difficult to distinguish from natural

cirrus clouds. Here we provide satellite-driven analysis of long-lived heat trapping by contrails over North and South America.
We aggregate a dataset of GOES-16 estimated outgoing longwave radiation and advected trace density of flight paths, and
apply causal inference to discern the effect of contrails while controlling for radiative and cloud confounders. As a means
of validation, we also generate synthetic datasets with known ground truth, and confirm that applying the causal inference
method is able to recover the synthetic ground truth. Since this method yields an estimate which has some differences from

both "instantaneous radiative forcing" ($iRF$) and "effective radiative forcing" ($ERF$) estimates which have been reported in
the literature so far, we introduce the new term "observational radiative forcing, 12 hours" ($oRF_{12}$). Our analysis estimates the
longwave $oRF_{12}$ from contrails over the Americas averaged 46.9 gigajoules per flight kilometer (95% CI: 35.8 to 58.0 GJ/km)
during April 2019 to April 2020.

## 1 Introduction

Condensation trails (contrails) are the ice clouds that form behind jet aircraft as they travel through sufficiently cold and
humid air (Schumann, 1996). When ambient atmospheric humidity is supersaturated with respect to ice, contrails can persist
for several hours and influence the Earth's energy budget by reflecting incoming solar radiation (during daytime only) and
trapping outgoing longwave radiation (at all hours). The net balance of these two effects has been estimated to be warming on
average and comparable in magnitude to the warming impact of aviation $CO_2$ emissions (Lee et al., 2021).

20       A small number of satellite based studies of contrail radiative forcing have been reported, and can be categorized as following
one of three different approaches, each with inherent limitations. The first approach fits a regression to determine the contribu-
tion of air traffic to top-of-atmosphere Outgoing Longwave Radiation (OLR) by restricting analysis to an observation region
in the North Atlantic, while controlling for confounders by including a regression term for a South Atlantic region that did not
contain air traffic (Schumann and Graf, 2013). This spatial restriction limits the potential for the approach to be extended to

a globally representative analysis of contrail forcing. The second category of approaches use a linear contrail detection mask
(Vázquez-Navarro et al., 2015; Bock and Burkhardt, 2016) or manual bounding of a clear-sky outbreak (Wang et al., 2024a), to





demarcate which pixels of the satellite imagery contain exclusively contrails. The reliance on such masks makes it challenging to estimate the overall radiative forcing of contrail cirrus, as masking becomes increasingly error prone in longer-lived contrail cirrus due to the difficulty of distinguishing naturally occurring cirrus apart from contrails which have evolved out of their

initial (distinctively linear) morphology. The third approach is (Schumann et al., 2021), which leverages the months in 2020 with COVID-19 disruptions in air traffic as a counterfactual, and subtracts off satellite estimates of OLR in those months from the prior year's estimated OLR values to estimate the aviation contribution to OLR. This approach is limited by the uniqueness of this air traffic disruption, and also faces signal/noise ratio issues due to the large natural inter-annual variation in weather conditions (Wilhelm et al., 2021).

Therefore to date, the only technique which could make estimates of the radiative forcing of globally representative samples of long-lived contrails has been parameterized models, for example climate simulations or microphysical models (Burkhardt and Kärcher, 2011; Chen and Gettelman, 2013; Schumann et al., 2015; Bickel et al., 2020; Teoh et al., 2024). These parameterized models have been developed in recent decades and have used observational data to inform their individual parameter selections, including passive satellite imagers, ground and space lidar, radiosonde, ground camera observations, cloud cham-

ber and in-situ ice crystal observations (Freudenthaler et al., 1995; Vázquez-Navarro et al., 2015; Schumann et al., 2017). However, the observational data constraining each individual parameter selection are gathered in specific locations and times which may not be representative of larger populations of contrails, especially considering the high inter-annual variance in contrail prevalence due to year over year meteorological variance (Wilhelm et al., 2021). In particular, microphysical contrails models have been shown to be sensitive to the numerical weather humidity fields they take as input (Teoh et al., 2024; Agarwal

et al., 2022), necessitating ongoing research into various correction methods including parametric scaling (Teoh et al., 2022), histogram matching (Platt et al., 2024), and neural networks (Wang et al., 2024b).

Here we introduce a new satellite based methodology for estimating the radiative forcing of long-lived contrail cirrus spanning the diurnal cycle, that does not rely on contrail masking nor on humidity fields from numeric weather data, and has the potential to be applied to globally representative samples of contrails. We pull from the causal inference literature and frame the

problem as an observational study estimating an average treatment effect: the treatment is aircraft passage, and the effect is the change in OLR. This framework is well suited for our context: it could be a substantial undertaking to perform a randomized controlled experiment with real aircraft at the scale needed for discerning significance of OLR difference. Causal inference framing also provides a principled statistical structure for isolating a specific causal effect (contrails) from a complex system with numerous confounding variables (meteorology). Causal inference methods have recently been applied to remote sensing

problems (Wimberly et al., 2009; Deines et al., 2019; Serra-Burriel et al., 2021; Demarchi et al., 2023; Fons et al., 2023), and the regression fitted by (Schumann and Graf, 2013) to estimate contrail forcing in the North Atlantic follows a typical causal inference recipe of controlling for confounders, but did not describe their method in causal inference terminology. The main differences between (Schumann and Graf, 2013) and this work are: 1) we introduce rasterized 'advected trace density' as the treatment field, allowing a kilometer scale pixel of a geostationary imager to be the unit of analysis rather than averaging data

over millions of square kilometers 2) we control for confounders using both numerical weather data and geostationary satellite



data products, 3) the analyzed domain is spatially larger and more representative of the total population of all contrails, 4) we investigate our causal inference modeling using synthetic datasets having known ground truth.

## 2 Methods

### 2.1 Causal inference regression overview

To estimate the effect of air traffic on OLR, we employ a causal inference framework (Imbens and Rubin, 2015; Pearl, 2009; Holland, 1986; Rubin, 1974). Our approach is designed to isolate the warming impact of contrails from other atmospheric phenomena by modeling the relationship between satellite-observed OLR and air traffic density. This is achieved by explicitly controlling for key environmental confounders, such as the pre-existing weather patterns modeled by ERA5 and the observed cloud state from GOES-16. This helps to disentangle the contrail effect from the influence these conditions may have on flight routing and the formation of natural clouds.

The core of our method is a regression model where the dependent variable is the OLR from the COllocated Irradiance Network (COIN; McCloskey et al. (2023)), a high-resolution flux dataset derived from GOES-16 satellite imagery. As detailed in Section 2.2.1, COIN provides OLR estimates at the high spatio-temporal resolution needed to observe contrail effects. This observed OLR is modeled as a linear function of three primary variables: 1) The OLR from the European Centre for Medium-Range Weather Forecasts (ECMWF) ERA5 reanalysis (Hersbach et al., 2020), 2) the Geostationary Operational Environmental Satellite (GOES-16) L2 cloud phase product, and 3) the advected trace density of air traffic (detailed in Section 2.2.2). The mathematical formulation of our model is presented in Model (1).

$$\mathbb{E}[\text{OLR}_{\text{COIN}}|CP, A, \text{OLR}_{\text{ERA5}}] = \alpha_0 \cdot \text{OLR}_{\text{ERA5}} + \sum_{j=0}^{4} I_{\{\text{CP}=j\}}(\beta_j + \gamma_j A). \tag{1}$$

In this model, $\text{OLR}_{\text{COIN}}$ and $\text{OLR}_{\text{ERA5}}$ are the OLR values from the COIN and ERA5 reanalysis, respectively, while $A$ represents the advected trace density of air traffic. The term $CP$ is the GOES-16 cloud phase, categorized as (0) clear sky, (1) liquid water, (2) supercooled liquid water, (3) mixed-phase, or (4) ice. Since contrails are ice clouds, they are included in the $CP = 4$ category. The model's parameters are interpreted as follows: $I_{\{\text{CP}=j\}}$ is an indicator function for cloud phase $j$; $\beta_j$ represents the baseline OLR adjustment for category $j$; and $\gamma_j$ is the conditional effect of advected trace density within that same category. We defer a detailed description of these variables to Section 2.2, and discuss the causal model next.

By incorporating ERA5 OLR as an independent variable, we control for the vast majority of natural atmospheric processes that influence OLR. Since the ERA5 model parameterizes clouds based on the thermodynamic state of the atmosphere and does not assimilate all-sky satellite radiances, it does not explicitly account for contrail formation (Hersbach et al., 2020). Consequently, the discrepancies between the COIN and ERA5 OLR estimates can be attributed to factors not captured in the ERA5 reanalysis, including the radiative forcing from contrails. This type of use of ERA5 as a counterfactual for analyzing radiative impacts has been applied in recent studies on aerosol-cloud interactions (Chen et al., 2022; Jia et al., 2024). Figure 1





shows the residuals from estimating the model $\mathbb{E}[\,\text{OLR}_{\text{COIN}}\mid \text{OLR}_{\text{ERA5}}\,] = \lambda_0 + \lambda_1\,\text{OLR}_{\text{ERA5}}$, linear contrails and the contrail cirrus they evolve into are clearly visible in the residual imagery, demonstrating the presence of the contrail signal in the data.

A key challenge, however, is that ERA5's representation of even natural clouds is not perfect. Errors in the model's prediction of cloud location, phase (e.g. ice vs. water), or optical properties can also create differences between the observed COIN OLR and the predicted ERA5 OLR. This is a source of confounding, as air traffic often occurs in the same upper-tropospheric regions that are prone to natural cirrus formation. Without further controls, the model might wrongly attribute the radiative effect of a natural cirrus cloud (that was simply missed or misrepresented by ERA5) to the presence of air traffic.

To address this, we include the GOES-16 L2 cloud phase product as an independent variable. This provides direct satellite observations of the actual cloud state (e.g., clear sky, ice cloud, water cloud) in each pixel. By including this variable, we allow the model to account for the systematic differences between COIN and ERA5 OLR that are due to the presence of different types of clouds independent of air traffic. For example, the model allows us to estimate the average OLR discrepancy that occurs when GOES-16 observes an ice cloud that ERA5 did not estimate to be there. Explicitly controlling for the observed cloud state allows the model to more accurately isolate the remaining effect attributable to the advected trace density. This step allows Model (1) to better distinguish between OLR anomalies caused by ERA5 inaccurately estimating cloud properties (e.g. by missplacing a cirrus cloud - represented by $\beta_4$) apart from an anomaly caused by a contrail that ERA5 does not estimate to exist there at all (represented by $\gamma_4$). The resulting coefficient, $\gamma_4$, therefore represents the average conditional effect of increasing advected trace density on OLR for all pixels classified as ice clouds (CP=4), a group which contains both natural cirrus and contrails. It can be interpreted as the mean effect of a contrail on top of any ice cloud scene.

The model attributes the difference between COIN and ERA5 OLR to the cloud phase and advected trace density. Through careful normalization of the advected trace density units, the fitted slope of this regression directly quantifies the average contrail effect on OLR per kilometer of flight, as illustrated in Figure 2.

The coefficients $\gamma_j$ from Model (1) represent the change in COIN OLR per unit of advected trace density for each cloud phase. To find the overall average effect, we calculate the Average Treatment Effect (ATE) by weighting each coefficient by the probability of that cloud phase occurring in the dataset. To give some intuition to this, consider an increase of advected trace density $A$ of $\delta$ for supercooled liquid water (cloud phase category 2) pixels with same $\text{OLR}_{\text{ERA5}}$ baseline is: $\mathbb{E}[\text{OLR}_{\text{COIN}}|CP = 2, A = a + \delta, \text{OLR}_{\text{ERA5}}] - \mathbb{E}[\text{OLR}_{\text{COIN}}|CP = 2, A = a, \text{OLR}_{\text{ERA5}}] = \delta\gamma_2$. Note that, according to our assumptions, we compare sets of pixels that are identical except for advected trace density, and we attribute the change in $\text{OLR}_{\text{COIN}}$ to advected trace density. Therefore, we can use (1) to estimate the average treatment effect (ATE) of an increase in advected trace density on $\text{OLR}_{\text{COIN}}$ by integrating out the ERA5 OLR baseline which yields:

$$\text{ATE} = \frac{1}{\delta}\left(\mathbb{E}[\text{OLR}_{\text{COIN}}|A = a + \delta] - \mathbb{E}[\text{OLR}_{\text{COIN}}|A = a]\right) = \sum_{j=0}^{4}\gamma_j P(CP = j), \tag{2}$$

where $\delta$ is the amount of change in advected trace density, $P(CP = j)$ is the probability that a GOES-16 pixel will have cloud phase category $j$, and $\mathbb{E}[\text{OLR}_{\text{COIN}}|A = a]$ is the expected $\text{OLR}_{\text{COIN}}$ conditional on advected trace density, in which $\text{OLR}_{\text{ERA5}}$ and cloud phase have been integrated out.





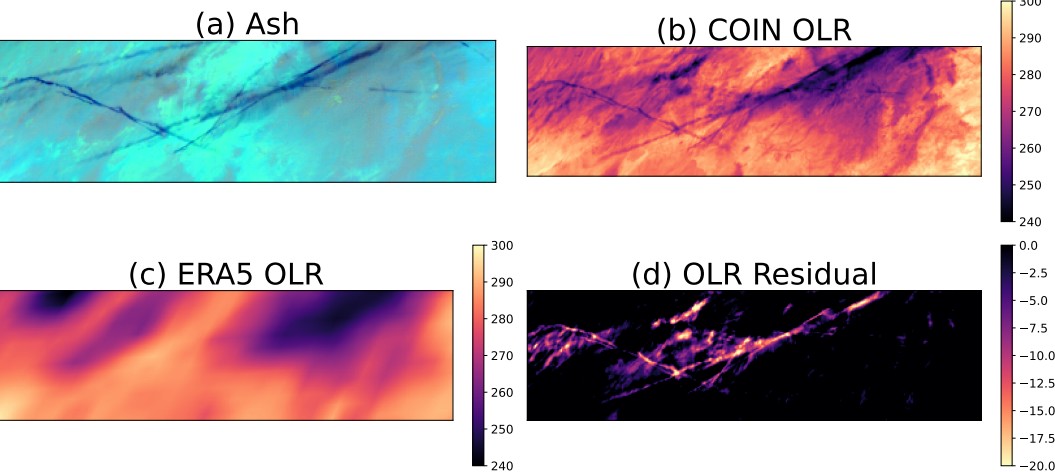

**Figure 1.** A contrail outbreak scene at 2019-07-18 14:20 UTC, over the southwestern United States. The GOES-16 ABI Ash longwave color scheme in panel (a) highlights the contrails as dark blue linear objects. In panel (b) the GOES-16 based COIN OLR field (in $W/m^2$) also shows the contrails trapping thermal radiation, but contrails cannot be seen in the panel (c) ERA5 OLR because ERA5 does not model nor assimilate contrails. The panel (d) OLR Residual image is generated by using ordinary least squares to predict COIN OLR as a function of ERA5 OLR and plotting the linear fit residuals; the bright, linear, and web-like features highlight young and aged contrail cirrus that are captured by the satellite observations but not present in the ERA5 model, demonstrating the presence of the contrail signal in the data. Note panel (d) clips the OLR residual to only show negative residuals, for visual clarity; the quantitative analyses – Models (1) and (3), and Equation (2) – use all values of their respective variables.

This equation defines the Average Treatment Effect (ATE) as the average change in $\text{OLR}_{\text{COIN}}$ per unit increase ($\delta$) in advected trace density ($A$). The difference in expectations on the left represents this definition, where the integration is marginalized over the population distributions of all other covariates ($\text{OLR}_{\text{ERA5}}$ and cloud phase). Due to the linear structure of our model, this simplifies to the expression on the right. This final expression is a weighted average of the conditional effects, where $\gamma_j$ is the specific effect of advected trace density on $\text{OLR}_{\text{COIN}}$ when a pixel is in cloud phase $j$, and $P(CP = j)$ is the area-weighted marginal probability of a pixel belonging to that cloud phase category. This weighting ensures that larger pixels away from the satellite's nadir contribute proportionally to the final estimate.

Analysis results are based on 183 days of data from April 2019 - April 2020 (every other day), in the region seen in Figure 5 which includes much of both North and South America.

## 2.2 Data fields used in causal regressions

Prior to performing regressions, data fields are reprojected (if necessary) onto a common spatial grid of pixels: the grid used by GOES-16 Advances Baseline Imager (ABI) longwave bands, which has shape 5424x5424 covering the western hemisphere as viewed from geostationary orbit at longitude -75 degrees. Hereafter this will be referred to as the "ABI pixel grid." The pixels





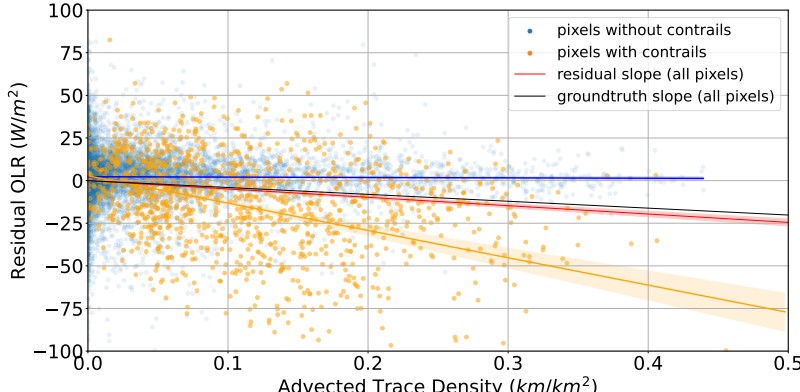

**Figure 2.** Relationship between advected trace density (x-axis) and the ordinary least squares residuals of COIN OLR as a function of ERA5 OLR (y-axis), on a sample of synthetic data with known ground truth (see Section 2.3 for further details). The plot shows how the regressed difference between ERA5 and COIN quantifies the average contrail effect on OLR. Colors code whether residuals come from contrail pixels (orange) or non-contrail pixels (blue). Fitted lines between flight density and residuals illustrate how the contrail effect is estimated separately when using contrail-only pixels (contrail effect) vs non-contrail (no effect) or all pixels (advected trace density effect from contrails).

are nominally 2km per side, but increase in size with increasing viewing angle away from nadir. Where calculations require multiplying by pixel area, we estimate the pixel area in $m^2$ by calculating the great circle distance between adjacent pixel centers and applying the trigonometric formulae for parallelogram area based on these side lengths. For visualization and area formula see Appendix B.

### 2.2.1 COllocated Irradiance Network (COIN) flux

The COIN model was developed as a method of estimating broadband irradiance (flux) from GOES-16 ABI narrowband radiances, in order to have the spatio-temporal resolution needed (nominally 2km spatial resolution with a 10minute refresh rate) to observe contrail effect on the Earth's top-of-atmosphere upwelling irradiance (McCloskey et al., 2023). Briefly, COIN is a neural network model trained to make pixel-wise estimates of flux (OLR and RSR in units of $W/m^2$) on a dataset of GOES-16 ABI input radiances collocated with Cloud and Earth's Radiant Energy System (CERES) Level2 SSF flux labels (Wielicki et al., 1996), whose error was minimized by backpropagating gradients through the CERES sensor's point spread function. Here we use the COIN estimates of OLR which were released alongside McCloskey et al. (2023) and available at gs://upwelling_irradiance/.

### 2.2.2 Advected trace density

We introduce the concept of advected trace density which represents the expected value of contrail length (in kilometers) inside an area, if every flight path had created a persistent contrail.





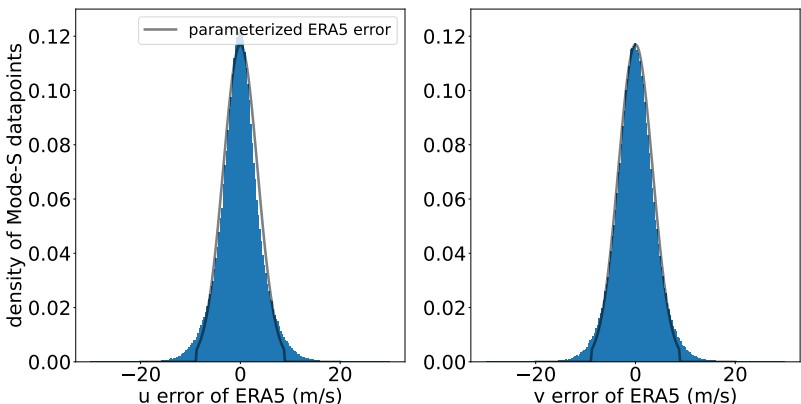

**Figure 3.** Distribution of ERA5 wind errors. The histograms show the difference between ERA5 u- and v-component winds and those calculated from aircraft Mode-S data. A Gaussian fit (black line) is used to model the wind error uncertainty, which grows at an estimated 12.4 km/h and is incorporated into the advected trace density calculation.

Advected trace density is an extension of the concept of advecting ADS-B flight trajectory waypoints, which are then used to derive collocation datasets with satellite sensor data (Duda et al., 2004; Tesche et al., 2016; Geraedts et al., 2024; Sarna et al., 2025); the name "advected trace" was introduced in Chevallier et al. (2023). We begin by advecting flight waypoints provided in ADS-B data licensed from flightaware.com and Aireon, following Geraedts et al. (2024). Flight waypoints are advected using ERA5 wind data with a Runge Kutta 3D method, and additionally sediment downwards using the terminal velocity of an estimated ice crystal size average. We advect each flight waypoint for 12 hours.

We extend the advection to model the uncertainty in wind error as a Gaussian whose standard deviation grows linearly over time. In particular, as shown in Fig 3 we found the error of the ERA5 $u$ and $v$ components of wind grow at 12.4 kilometers per hour, when compared against Mode-S computed wind speeds licensed from flightaware.com. We also then normalize the advected trace density field, so that when we rasterize it into the ABI pixel grid, it has units of flight length divided by pixel area: $km/km^2$. An example of its rasterization can be seen in Fig 4.

In its simplest form, advected trace density is a 2-dimensional spatial field ('x' and 'y' pixel coordinates) on the ABI pixel grid, that also varies with time. To analyze the effect of contrail age, we define a set of cumulative advected trace density variables, $A_H$, for $H = 0, \ldots, 11$. Each variable $A_H$ represents the total advected trace density from all flight paths with an advection age of less than $H + 1$ hours. For example, $A_0$ includes the density from flight traces advected for less than one hour, $A_1$ includes the density from all advected traces younger than two hours, etc. This formulation allows us to fit a separate regression model for each cumulative age $H$, as shown in Model (3), in order to estimate the total forcing as a function of contrail lifespan.

$$E[\mathrm{OLR_{COIN}}|\mathrm{CP}, A_H, \mathrm{OLR_{ERA5}}] = \alpha_{0H} \cdot \mathrm{OLR_{ERA5}} + \sum_{j=0}^{4} I_{\{\mathrm{CP}=j\}}(\beta_{jH} + \gamma_{jH}A_H). \tag{3}$$





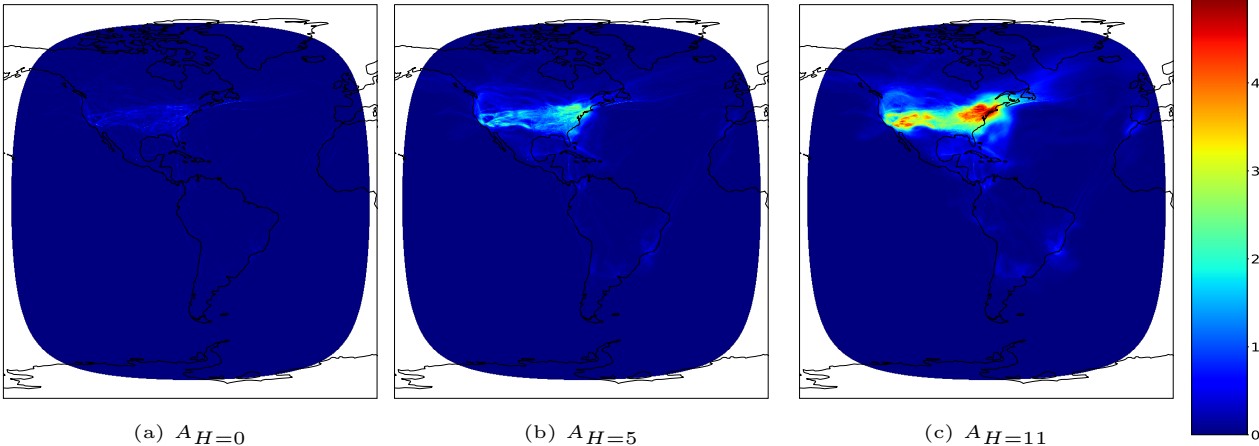

(a) $A_{H=0}$       (b) $A_{H=5}$       (c) $A_{H=11}$

**Figure 4.** An example of the rasterized advected trace density field (in units of flight kilometer per square kilometer, $km/km^2$) from May 14, 2019 at 01:00:00 UTC, advected for up to 1 hour ($A_{H=0}$), up to 6 hours ($A_{H=5}$) and up to 12 hours ($A_{H=11}$).

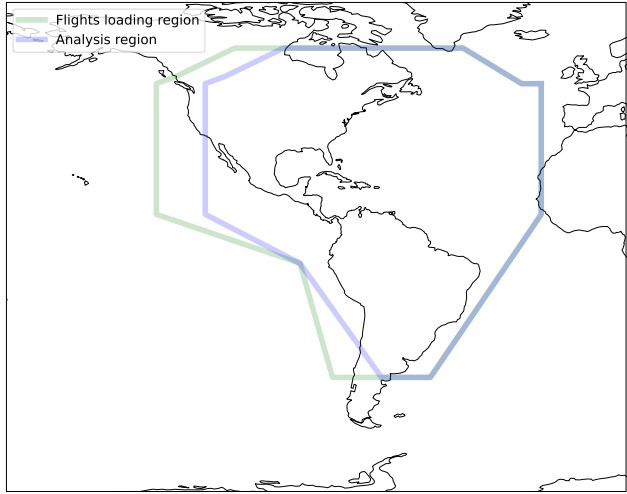

**Figure 5.** Geographic domain of the study. ADS-B flight data was loaded from the larger (green) outer polygon to create the advected trace density field. To mitigate advection related edge effects, the causal regression was then performed on pixels sampled from within the inner blue polygon which is smaller in places that have prevailing flight level winds in a certain direction.

Note that we fit Model (3) separately for each cumulative age: $H = 0, \ldots, 11$. To mitigate spatio-temporal edge effects from advection in the causal regression, we load ADS-B waypoints from a larger bounding loop (Figure 5, "Flights loading region") than we sample from when fitting Model (1) (Figure 5, "Analysis region"). For each of the 183 days in the analysis, we load
175 (and perform 12hr advections) for ADS-B waypoints from 36 hour windows, and then only sample from the final 24 hours of each 36-hour window when estimating the causal model.





### 2.2.3 ERA5 OLR

Prior to performing the estimation of the causal model parameters in Model (1), we must convert the ECWMF ERA5 OLR field (which has units $J/m^2$ accumulated over the hour timestep of the weather model, on a 0.25 degree latitude/longitude grid) into a form that allows pixel-wise analysis with the other data fields in the model. To accomplish this, we first divide the hourly-accumulated ERA5 OLR by 3600 seconds to yield flux in units of $W/m^2$, and then apply a reprojection using linear interpolation in the spatial dimension onto the ABI pixel grid, applying parallax correction for the top of atmosphere flux using a nominal top-of-atmosphere altitude of 20km. Temporally, we apply nearest-neighbor interpolation.

### 2.2.4 GOES-16 L2 Cloud Top Phase product

The GOES-16 ABI Cloud Top Phase algorithm (Heidinger et al., 2020) determines the top-altitude cloud phase by analyzing infrared radiances (specifically $7.4\mu m$, $8.5\mu m$, $11\mu m$, and $12\mu m$ channels). It converts these radiances into effective cloud emissivities and "beta-ratios" (ratios of effective absorption optical depths), and applies a decision tree based on threshold tests to classify cloud tops into categories of warm liquid water, supercooled liquid water, mixed phase, and ice.

### 2.2.5 Block Bootstrap uncertainty quantification

To estimate confidence intervals of our central estimate of contrail longwave RF, we apply block bootstrap (Künsch, 1989) where a block consists of pixels within the same day. This helps maintain the flight traffic and synoptic weather covariance structure of the data. Specifically, we first sample 183 days with replacement, and then proceed to sample pixels within each sampled day to sample a total of 500,000 pixels. We then estimate the model coefficients and compute the corresponding value in units of gigajoules per flight kilometer. This sampling and estimation process is repeated 1,000 times. We show the central estimate derived from a sample of 3 million pixels, along with the 2.5th and 97.5th block-bootstrap percentiles as the 95% confidence interval.

## 2.3 Synthetic dataset validations

To validate and probe this method, we generate a few synthetic versions of this type of dataset using the pycontrails v52.2 implementation (Shapiro et al.) of CoCiP (Schumann, 2012), where we are able to record the synthetic known ground truth value of the longwave forcing per flight kilometer to confirm the regression method can recover it from being embedded in situ among natural confounders and error covariances. We created the synthetic datasets exclusively in the southern hemisphere, where flight density is much lower than in the northern hemisphere, to avoid accidentally creating unrealistic modeling difficulties by aligning synthetic contrails with real contrails. The synthetic dataset is composed of the same data fields as the real dataset:

1. A synthetic advected trace density field is rasterized, by loading real ADS-B waypoints for flights traversing the Conterminous United States (CONUS), flipping their latitudes into the southern hemisphere (multiplying by -1), and then



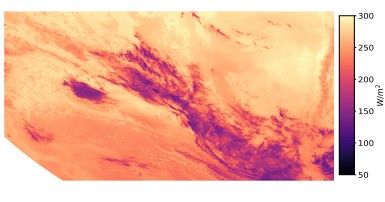

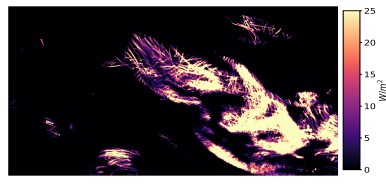

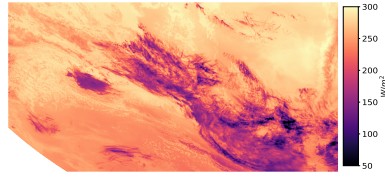

(a) Observed COIN OLR            (b) CoCiP longwave forcing            (c) Synthetic COIN OLR

**Figure 6.** Example of a synthetic COIN OLR field used for model validation. (a) shows the observed OLR field in the southern hemisphere. The observed flight tracks from the northern hemisphere were latitude-flipped and used to generate contrails according to the CoCiP model as shown in (b). The radiative forcing from these synthetic contrails was then subtracted from the observed OLR field in the southern hemisphere, creating a dataset with a known ground truth forcing as shown in (c).

applying the same 12hr advection and rasterization as described in Sect. 2.2.2; advections were performed using the ERA5 wind data from their latitude-flipped (southern hemisphere) waypoint locations.

2. A synthetic COIN OLR field is rasterized, starting from the real COIN field provided for the southern hemisphere by (McCloskey et al., 2023). Then, taking the same latitude-flipped ADS-B waypoints used above, we estimate the contrail longwave forcing using CoCiP; in all pixels where CoCiP estimates that (based on the unmodified southern hemisphere ERA5 weather fields) there would be nonzero forcing we subtract it from the original COIN OLR at that location on the ABI pixel grid. The details for rasterizing the CoCiP forcing follow the rasterization of CoCiP contrail opacity in (Sarna et al., 2025), with the exception that instead of rasterizing a thresholded opacity mask here we are rasterizing the CoCiP 'rf_lw' property (or in one variant below, the flux as a parameterization of the 'tau' optical depth property). An example of the synthetic COIN OLR field ("Linear overlap" variation) can be seen in Fig 6 (c).

3. A synthetic GOES-16 L2 Cloud Top Phase field is used, which is only a slightly modified version of the real data: pixels where CoCiP rf_lw is nonzero have a 74% random chance of becoming set to Cloud Phase 4 (ice cloud), to account for inaccuracies reported in (Jiménez, 2020).

4. An unmodified ERA5 OLR field (the real field from the southern hemisphere, reprojected as in Sect 2.2.3) is used in the estimation of Model (1).

We generate four variations of such synthetic datasets:

– "Linear overlap": in this variation, CoCiP estimates of rf_lw are summed linearly in rasterized pixels (This is currently the default operating mode of pycontrails and all CoCiP studies in the literature we are aware of to date).

– "Sublinear overlap": in this variation, the CoCiP estimates of tau (contrail optical depth) from different contrails in the same pixel are summed in log space, creating an "effective tau" which is then converted to rf_lw via an approximation detailed in Appendix A. This variation investigates whether estimating Model (1) as a linear function introduces





estimation error when the effect on OLR with increasing advected trace density is likely to be somewhat sublinear in reality.

- – "Measurement bias": in this variation, each of a "Linear overlap" and "Sublinear overlap" synthetic dataset are further modified to introduce simulated measurement bias of the type that was noted as occurring in the COIN estimates relative to CERES flux labels in Figure 4 of (McCloskey et al., 2023). This variation investigates whether the estimation of Model (1) is robust to this type of systematic bias.

- – "Null": in this variation, we fit Model (1) using a latitude-flipped synthetic advected trace density field as usual, but coupled with an unmodified real COIN OLR field from the southern hemisphere which did not have aircraft in those places making contrails. That is, in this variant the ground truth contrail forcing caused by this advected trace density is zero, but typical levels of discrepancy between ERA5 OLR and COIN OLR from other causes exist in the dataset and could potentially bias the regression estimate to return a non-zero answer; using this variant, we confirm that the Model (1) regression correctly returns an estimate of zero contrail forcing.

## 3 Results

### 3.1 Insight from synthetic datasets

An important observation from regressing against the synthetic OLR datasets is that many of the discrepancies between COIN OLR and ERA5 OLR are of a similar (or larger) magnitude than the effect size of contrail forcing that has been estimated in the literature to date. Because of this—and because air traffic levels are high enough that large portions of CONUS have nonzero advected trace density at most hours of the day—it is likely that at any given time there exists co-occurring advected trace density and COIN-ERA5 discrepancy in the same pixels not actually caused by contrails. In this situation, a small sample of such pixels could return a non-zero radiative forcing estimate which is entirely caused by such spurious correlations. For our purpose of making a low bias estimate of average contrail longwave forcing, we must draw a large sample of independent observations so these spurious correlations can cancel each other out during the regression. Crucially, we note that spatio-temporally adjacent pixels are *not* independent observations: to usefully increase the sample size pixels must be sampled from a large number of independent synoptic weather patterns, and we achieve this by sampling from different days. See Fig 7 for details on estimate bias as a function of number of days sampled and number of pixels sampled; we observe that only a few hundred thousand pixels are necessary for a low bias estimate of the synthetic ground truth forcing, as long as they are sampled from a few dozen different days.

As seen in Fig 8, we find that with a sufficiently sized (and sufficiently independent) input sample, the regression method provides a low bias estimate for all of the "Linear overlap", "Sublinear overlap" and "Measurement bias" tests, as well as correctly returning close to zero contrail forcing in the "Null" test .





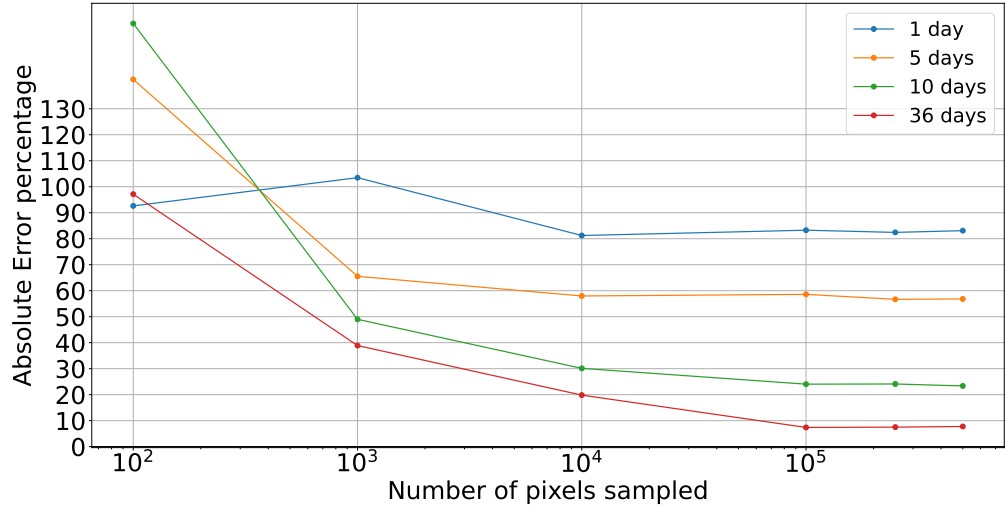

**Figure 7.** Convergence of the longwave forcing estimate as a function of the number of pixels sampled in the "Linear overlap" synthetic dataset test. As the number of independent days sampled increases the error decreases, but sampling more than a few hundred thousand pixels does not seem to improve the error.

We also use the synthetic dataset to assess the calibration of our bootstrap confidence intervals, and indeed saw that on the "Linear overlap" dataset the nominal 95% confidence intervals correctly captured the known ground truth in 95% of trials. This
260  confirmed that our chosen day-block bootstrap method is well-calibrated.

## 3.2 Central estimate of contrail longwave forcing in the Americas

In the spatial domain seen in Fig 5(b), during Apr2019 - Apr2020, contrails averaged 46.9 gigajoules of longwave $oRF_{12}$ per flight kilometer. The 95th percentile confidence interval is 35.8 to 58.0 GJ/km (p<0.001). We also performed a permutation test, randomly permuting the advected trace density among pixels in the dataset but leaving the other fields unmodified, followed by
265  fitting Model (1) on the permuted version of the dataset. If this test returned a nonzero answer, it could indicate that our central estimate may not be statistically significant. However, the permuted regression correctly returns very close to zero forcing: estimate: -0.01 (95% CI: -0.47 to 0.46) GJ/km.

To place our observational result in the context of an established contrail model, we also calculated the instantaneous radiative forcing (iRF) for the same set of flights using CoCiP (Shapiro et al.; Schumann, 2012). Running CoCiP over the same time
270  period (Apr2019 - Apr2020) and geographic region shown in Fig 5 yielded a total longwave iRF estimate of 60.7 gigajoules per kilometer. Note that as is currently common in the literature we executed CoCiP in "offline" mode where it is not coupled to the simulated atmosphere it uses as input (in this case ERA5), so it does not include any atmospheric feedback effects which may be captured by our observational method.





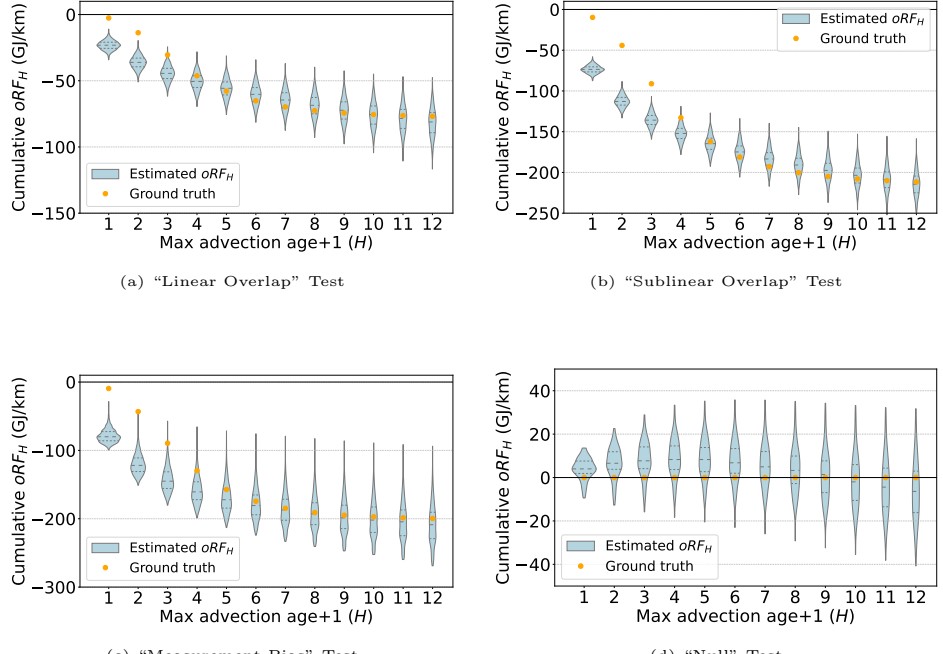

(a) "Linear Overlap" Test         (b) "Sublinear Overlap" Test

(c) "Measurement Bias" Test         (d) "Null" Test

**Figure 8.** Validation of the causal inference method on synthetic datasets. The plots show the estimated contrail forcing (blue violins) versus the known ground truth forcing (orange dots) as a function of contrail age using Model (3). Each panel shows the distribution of the estimated forcing ($oRF_H$) from the twelve separate OLS fits, one violin per fit of a cumulative age $H \in \{0, \ldots, 11\}$. The model successfully recovers the known ground truth in the (a) "Linear Overlap", (b) "Sublinear Overlap", and (c) "Measurement Bias" tests, and (d) correctly returns a near-zero forcing for the "Null" test, where no contrails were present.

The results from the fitted Model 1 are presented in Table 1. The primary finding is the Average Treatment Effect (ATE),
275   which shows that an increase in advected trace density has a statistically significant warming effect, changing OLR by an average of -13.04 $\frac{W/m^2}{km/km^2}$ (95% CI [-16.21, -9.87]). The units $\frac{W/m^2}{km/km^2}$ reflect that this number is the slope of the causal linear regression: the change in OLR ($W/m^2$) per change in advected trace density ($km/km^2$). To convert the resulting units into being normalized only by the flight kilometers, we note the denominators of each quantity are both an area and can be canceled out by multiplying the $m^2$ by 1e6 to convert it to $km^2$, leading to an intermediate unit of $W/km$. Then to convert the numerator
280   Watts to Joules, it is multiplied by 3600 seconds in an hour. Finally converting to Gigajoules per flight kilometer we divide by 1e9:

$$\frac{W/m^2}{km/km^2} = \frac{1,000,000 W/km^2}{km/km^2} = \frac{1,000,000 W}{km} = \frac{3,600,000,000 J}{km} = \frac{3.6 GJ}{km}. \qquad (4)$$

This sequence of unit conversion steps is therefore equivalent to multiplying by 3.6 to yield our central estimate:



**Table 1.** Estimated Coefficients for Model 1, with Block Bootstrap Standard Errors and Confidence Intervals.

| Parameter | Feature | Estimate | 95% CI | SE |
|---|---|---|---|---|
| $\hat{\alpha}_0$ | ERA5 OLR (unitless) | 0.591 | [0.581, 0.601] | 0.005 |
| $\hat{\beta}_0$ | cloud phase (CP) 0: clear sky ($W/m^2$) | 116.288 | [113.307, 119.269] | 1.521 |
| $\hat{\beta}_1$ | CP 1: warm liquid water ($W/m^2$) | 112.254 | [109.523, 114.985] | 1.393 |
| $\hat{\beta}_2$ | CP 2: supercooled liquid water ($W/m^2$) | 88.669 | [86.167, 91.172] | 1.277 |
| $\hat{\beta}_3$ | CP 3: mixed ($W/m^2$) | 79.238 | [76.805, 81.672] | 1.242 |
| $\hat{\beta}_4$ | CP 4: ice clouds ($W/m^2$) | 76.410 | [74.080, 78.739] | 1.188 |
| $\hat{\gamma}_0$ | CP 0: clear sky × Advected trace density ($\frac{W/m^2}{km/km^2}$) | −29.305 | [-34.915, -23.696] | 2.862 |
| $\hat{\gamma}_1$ | CP 1: warm liquid water × Advected trace density ($\frac{W/m^2}{km/km^2}$) | −7.517 | [-11.169, -3.864] | 1.863 |
| $\hat{\gamma}_2$ | CP 2: supercooled liquid water × Advected trace density ($\frac{W/m^2}{km/km^2}$) | −3.670 | [-6.924, -0.417] | 1.660 |
| $\hat{\gamma}_3$ | CP 3: mixed × Advected trace density ($\frac{W/m^2}{km/km^2}$) | −6.369 | [-12.675, -0.063] | 3.217 |
| $\hat{\gamma}_4$ | CP 4: ice clouds × Advected trace density ($\frac{W/m^2}{km/km^2}$) | −4.304 | [-8.346, -0.262] | 2.062 |
| $\sum_{j=0}^{4} \hat{\gamma}_j \hat{P}(CP=j)$ | Average Treatment Effect ($\frac{W/m^2}{km/km^2}$) | −13.040 | [ -16.21, -9.87] | 2.128 |

$$-13.04 \, \frac{W/m^2}{km/km^2} \cdot 3.6 = -46.9 \, GJ/km \quad [-35.8, -58.0]. \tag{5}$$

Note that the sign being negative here is intentional: the regression slope being negative indicates heat is being trapped (i.e. when OLR is smaller, not as much thermal radiation is escaping into space). Elsewhere in this work we report our central estimate $46.9 \, GJ/km$ consistent with the typical sign convention of earth system studies where positive numbers are warming.

Beyond the average treatment effect, one could hope to glean insights from the individual fitted coefficients of Table 1. The coefficient for the baseline ERA5 OLR ($\alpha_0 \approx 0.59$) shows an (expected) strong positive correlation with the COIN OLR. The fact that the estimate is substantially less than 1.0 indicates there remains a systematic scaling difference between the two OLR products even after the model accounts for the observed cloud states identified by GOES-16 L2 Cloud Phase product. The clear sky interaction with advected trace density is perhaps surprisingly the largest magnitude of the $\hat{\gamma}_j$ coefficients; one might expect the $\hat{\gamma}_3$ mixed phase and $\hat{\gamma}_4$ ice clouds to be largest, if the underlying cloud phase data were accurately identifying the top phase as ice in the cloudy pixels that contain contrail forcing. However, (Jiménez, 2020) analyzed the GOES-16 Cloud Mask product accuracy compared to lidar ground truth measurements from CALIPSO and found that overall the clear sky detection accuracy was only 74.8%. Figure 8 from (Jiménez, 2020) in particular shows that in the winter in some higher latitudes (where contrail formation rates are expected to be relatively high due to lower temperatures coinciding with large amounts of flight traffic in the northern half of the United States) the clear sky detection accuracy can be as low as 35%, providing a plausible explanation for the large magnitude of $\hat{\gamma}_0$. Considering the (possibly unexpected) nonzero values of $\hat{\gamma}_1$ and $\hat{\gamma}_2$ for water clouds, we note that again uncertainties in the Cloud Phase classification are a likely explanation: contrail cirrus optical depth only



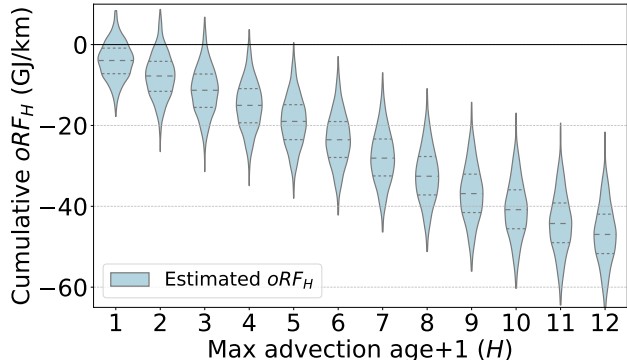

**Figure 9.** The estimated cumulative longwave forcing from contrails over the Americas as a function of maximum contrail age ($H$). Each violin in the plot shows a distribution of the estimated forcing ($oRF_H$) from one of the twelve separate OLS fits, one for each cumulative age $H \in \{0, \ldots, 11\}$. The curve shows that contrails continue to exert a warming influence for many hours after formation, with the total forcing increasing up to at least 12 hours. The central estimate at 12 hours is 46.9 GJ/km.

rarely exceeds 1.0 (Kärcher et al., 2009) and the validations performed by (Pavolonis, 2020) indicate the accuracy of "Optically Thin" or "Multilayered Ice" cloud classification is relatively poor, ranging from 39-58%.

### 3.3 Estimation of contrail lifespan

We utilize the advection age dimension of the advected trace density field to analyze how much forcing is associated with each advection age; we fit the 12 models from Equation (3), each using as the treatment only the advected trace density which was younger than hour $H$ (where $H < 12$), estimating the radiative forcing occurring as a result of air traffic less than $H$ hours after aircraft passage. The curve generated in this manner is shown in Fig 9. If we saw a flat horizontal line at age $n$ on the x-axis, it would indicate that no additional forcing was observable after those $n$ hours. However, we in fact see that the curve may continue downward even after 12 hours. This may indicate the contrail lifespan is longer than 12 hours.

### 3.4 Diurnal cycle

By performing regressions on pixels at times that are limited to a particular local hour of the day, we can observe the diurnal cycle of contrail forcing. The local hour timezone offset from the UTC time of the observation of the pixel is approximated as a function of the longitude of the pixel (each 15 degrees of longitude are grouped as a one hour timezone). We then perform 24 regressions fitting Model (1), and compare the resulting GJ/km values to the CoCiP estimated forcing rasterized in those same pixels in Fig 10. Note the x-axis is 48 hours long because the same 24 results have been concatenated back to back to give a more intuitive visualization of the trend lines.




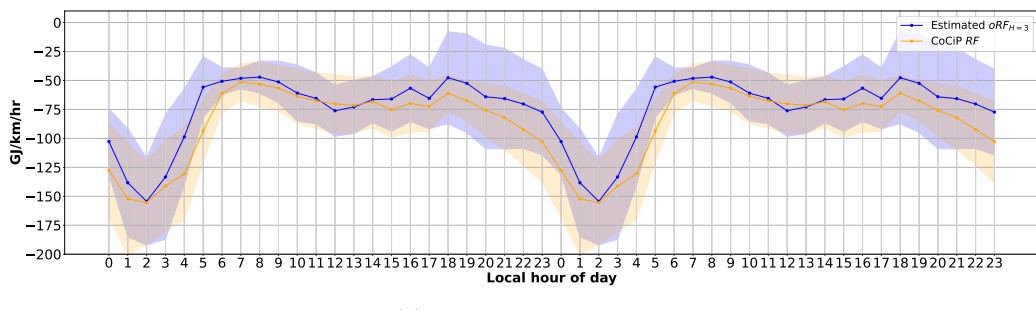

(a) "Linear Overlap" Test

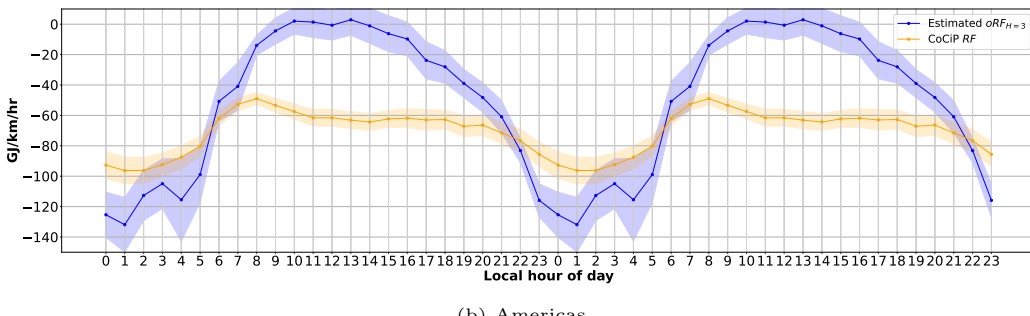

(b) Americas

**Figure 10.** The diurnal cycle of contrail longwave forcing. The plot shows the estimated forcing (GJ/km) by CoCiP and causal inference regression ($oRF_{H=3}$) for each local hour of the day, for (a) the "Linear Overlap" synthetic test and for (b) real contrail forcing estimated over the Americas. Note the flight km used to normalize all estimates is from advected trace density up to at most 3 hours old, to allow a temporally localized comparison. This illustrates how the longwave warming of contrails varies between day and night, likely due to interactions with the solar cycle and diurnal patterns in underlying cloud cover and air traffic. We note that in panel (b) the $oRF_3$ estimate differs from CoCiP's estimate primarily in the local mid-day and afternoon; this may be due to uncontrolled confounding or an atmospheric dehydration effect that is not modeled by CoCiP.

## 4 Discussion

### 4.1 Contextualizing the Forcing Estimate

We have provided here comparisons with CoCiP estimates of instantaneous radiative forcing (iRF), but the method we use here is not in fact estimating the exact same quantity. CoCiP estimates reported in this work (and in almost all of the literature to date) are executed in an "offline" mode, where the contrails modeled by CoCiP do not interact with the simulated atmosphere (as instantiated here in the ERA5 numeric weather data). Because of this, offline CoCiP estimates do not include feedback effects such as dehydration of the upper troposphere. These feedback effects have been noted to decrease the efficacy of contrail radiative forcing, and so climate simulations which take them into account have been reported as "Effective Radiative





Forcing" (ERF). (Lee et al., 2021; Bickel et al., 2025) Because our method is driven by satellite observations of the real Earth's atmosphere, arguably it contains the first 12 hours of such feedback effects, and so is not strictly an instantaneous radiative forcing. Considering one of the reported largest magnitude feedback effects (dehydration at flight levels, where decreased humidity — entrained into ice crystals of contrail cirrus and sedimented to lower altitudes — then decreases formation rates of natural cirrus), we do expect that some fraction of it is accounted for in the contrail forcing estimates reported in this work.

It may be the case that such a flight-level dehydration effect is responsible for differences between the $oRF_3$ estimate and the "offline" CoCiP estimate seen in Fig 10 (b). In support of that explanation, we note the diurnal timing is fairly consistent with the timing of the drop in linear contrail coverage sensitivity seen in Figure 2 of (Meijer et al., 2022). Alternately (or perhaps in combination with that effect), the discrepancies in Fig 10 (b) might be caused by unmeasured confounding, in particular aircraft systematically avoiding mid-day convective clouds (Honomichl et al., 2013; Guan et al., 2001). Given that in the synthetic

testbed shown in Fig 10 (a) the causal inference method is able to successfully recover the known synthetic ground truth diurnal curve, we conclude the explanation for the discrepancies in Fig 10 (b) is likely to lie in some remaining difference(s) between the synthetic testbed data and the data derived from real aircraft trajectories in the observed earth atmosphere. To distinguish which effects contribute to which degree, future work might explore even more realistic synthetic test data that is specifically designed to include more realistic correlations of aircraft trajectories with respect to natural clouds.

While 12 hour estimates we provide here are unlikely to account for any Nitrous Oxide (NOx) driven feedback effects, since they are expected to require multiple days of atmospheric chemistry reactions to become evident (Brasseur et al., 2016), it has not currently been reported what fraction of the full dehydration feedback effect is captured by $oRF_{12}$ estimates. Considering that our reported forcing is not quite iRF and not quite ERF, we propose for clarity of comparisons between contrail forcing estimate types that observation-driven analysis of contrail forcing can adopt the term "Observational Radiative Forcing" with

a subscript of the number of hours traced in the observations, e.g. this work reports $oRF_{12}$.

    There has been one report in the literature of CoCiP estimates being executed in an "online" mode, where it is coupled with the simulated atmosphere: (Schumann et al., 2015) reports a 15% reduction in efficacy of contrail radiative forcing due to the deyhdration effect. Taking this effect into account and applying a 15% reduction to CoCiP's offline estimate of 60.7 GJ/km on the analysis region from this work puts the CoCiP estimate within our observation-based estimate's confidence interval (35.8 to

58.0 GJ/km).

    The average lifespan we estimate of 12 or longer hours is a somewhat longer average lifespan than expected based on the CoCiP simulations of these same flights. It's possible this is due to CoCiP only modeling the fall streak portion of the contrail and not the smaller ice crystals of the contrail which sediment more slowly or not at all, as hypothesized recently by (Akhtar Martínez et al., 2025). It may also in part be an artifact of our analysis: as seen in Fig 8 panels (a), (b) and (c) the

blue "Estimated $oRF_H$" curves do not align perfectly with their respective orange synthetic "Ground truth" curves. We suspect the misalignment is due to correlations across adjacent hours $H$ of the advected trace density, where the contrail forcing from any given hour $H$ is also strongly correlated with adjacent hours and the causal model has no available counterfactual data to discriminate which hour it should be attributed to. It's possible that a causal model which explicitly tracks temporal state changes such as was used in (Fons et al., 2023) may yield an improved lifespan estimate.





## 4.2 Limitations and Future Work


While our model controls for large-scale meteorology with ERA5 and the observed cloud state with the GOES-16 Cloud Top Phase product, in the future it would be ideal to control with observational data that accounts for multiple cloud layers such as the GOES-R Cloud Cover Layer (CCL) product (Li, 2023); it is only available starting in May 2023 and does not coincide with the timespans when we have purchased satellite ADS-B data that is crucial for generating accurate advected trace density

over ocean regions. Besides multilayer observational data, future analysis would ideally also include observational data with improved accuracy for optically thin ice clouds; machine-learning approaches such as (Kox et al., 2014) that are specifically developed for ice clouds hold the most promise in our view.

Additionally there is a potential source of unmeasured confounding that remains, in the non-random nature of flight routing. Aircraft systematically avoid turbulent regions for safety, and the atmospheric dynamics that generate this turbulence might

have an association with natural cirrus clouds. Although including the GOES-16 Cloud Top Phase product in Model (1) partially accounts for this, the relationship between turbulence and observed cloud cover is not perfectly deterministic. Therefore, a subtle bias may be introduced, as the model might incorrectly attribute the radiative effects of these systematically avoided cloudy regions to the absence of air traffic. Future work could aim to address this more explicitly by incorporating turbulence forecast data as an additional predictor.

We also anticipate future refinements in advected trace density: while our current model assumes a linear growth in wind error uncertainty, a more sophisticated state-dependent error model could be developed where the error varies with geographic location, altitude, and local meteorological conditions like wind shear. Another advancement of wind uncertainty might be to calculate per-waypoint uncertainty based on performing advections in multiple weather ensemble members as done by (Meijer, 2024). Such improvements could reduce the spatial uncertainty of aged contrails and could sharpen the resulting

forcing estimates, particularly for contrails with longer lifetimes.

Our introduction of $oRF_{12}$ motivates targeted climate model experiments to better connect observational results with long-term climate impacts. By running climate simulations for 12 hours, one could quantify the forcing after only rapid atmospheric adjustments from this time interval have occurred. This may allow computing the ratio between iRF and $oRF_{12}$, which could then be compared to reported iRF/ERF ratios. This could illustrate how much of the total adjustment from iRF to ERF occurs

within the first 12 hours and the degree to which short-term observational metrics like $oRF_{12}$ can constrain the full climate response.

A crucial future extension of this methodology is to analyze the shortwave radiative effects of contrails to determine their net radiative impact. This requires quantifying the cooling effect from reflected solar radiation (RSR), which presents a more difficult challenge than the longwave analysis in this study: CoCiP and climate model estimates suggest contrail RSR signal will

be smaller in magnitude than the longwave forcing, and it is embedded within a much larger background variance; see Figure 3 in (McCloskey et al., 2023) for typical ranges of OLR and RSR. Additionally, RSR has a few more potential confounders than OLR: diurnal and annual cycles of solar illumination, stronger dependence on viewing angle including sunglint from water surfaces, and stronger dependence on surface type and potentially on cloud aerosol interactions. Therefore it may be required



to reduce wind uncertainty to strengthen the correlations of the treatment (advected trace density) with the outcome (change in

RSR), and thorough feature engineering of confounder inputs in regression models is likely needed.

An important extension of this work is to apply this analysis to global satellite observations; the current study is limited to the Americas, and its air traffic patterns and meteorological conditions may not be representative of global aviation. Similarly, given the high inter-annual variation reported for contrails, analyzing data from more years will be valuable. Additionally this framework can be extended to perform detailed subgroup analyses beyond the fleet-wide average: by partitioning the dataset

by engine type, local time of day, geographic region, or specific synoptic weather patterns, we could identify which flight and weather categories contribute disproportionately to contrail warming.

## 5 Conclusions

In this work, we provided a large-scale, observation-driven estimate of long-lived contrail radiative forcing. Our approach, grounded in a causal inference framework, successfully isolates the contrail longwave radiative forcing signal from confound-

ing weather effects by combining high-resolution satellite observations with flight data, thereby avoiding the limitations of traditional linear contrail masks. Our analysis quantifies the longwave 12-hour observational radiative forcing ($oRF_{12}$) to be 46.9 GJ/km and suggests that average contrail longwave radiative impact may exceed 12 hours. This work provides a new way for observationally estimating an important component of aviation's non-$CO_2$ impact and demonstrates an approach that could be used to validate atmospheric models and potentially evaluate future mitigation strategies.





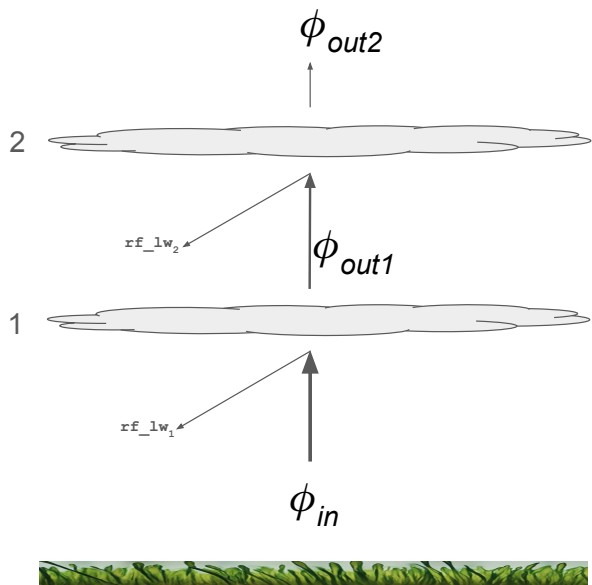

**Figure A1.** Sublinear overlap of contrail radiative forcing used in one synthetic validation test. $\phi_{in}$ is the initial upwelling irradiance, here estimated by COIN in $W/m^2$. After contrail 1 has attenuated the flux it becomes $\phi_{out_1}$ and after attenuation from contrail 2 it becomes $\phi_{out_2}$, which in the case shown would be the 'satellite-observed' synthetic OLR since contrail 2 is the highest cloud.

## Appendix A: Sublinear overlap formula

We generate a variation of a synthetic validation test to investigate whether estimations using a linear function introduces estimation error when the effect on OLR with overlapping contrails is likely to be somewhat sublinear in reality. To do this we generate synthetic data using a sublinear overlap function in pixels where multiple contrails from distinct aircraft are estimated by CoCiP. In typical CoCiP estimations of contrail forcing, the RF from these separate contrails would be linearly summed. Here instead we approximate the contrail RF effect via the sum of the optical depths.

The diagram Fig A1 shows two distinct contrail layers, having optical depth $tau_1$ and $tau_2$ respectively. Given that optical depth is defined as:

$$\tau = \ln(\frac{\phi_{in}}{\phi_{out}}),$$

To calculate the top level OLR ($\phi_{out_2}$) we can see:

$$\phi_{out_2} = \frac{\phi_{out_1}}{e^{\tau_2}} = \frac{\frac{\phi_{in}}{e^{\tau_1}}}{e^{\tau_2}} = \frac{\phi_{in}}{e^{\tau_1}e^{\tau_2}} = \frac{\phi_{in}}{e^{(\tau_1+\tau_2)}},$$

which in the general case of $n$ distinct contrails in the same pixel results in the formula we use to generate the "Sublinear overlap" variant of the synthetic test data:

$$\phi_{out_n} = \frac{\phi_{in}}{e^{(\sum_{i=1}^{n}\tau_i)}}. \tag{A1}$$




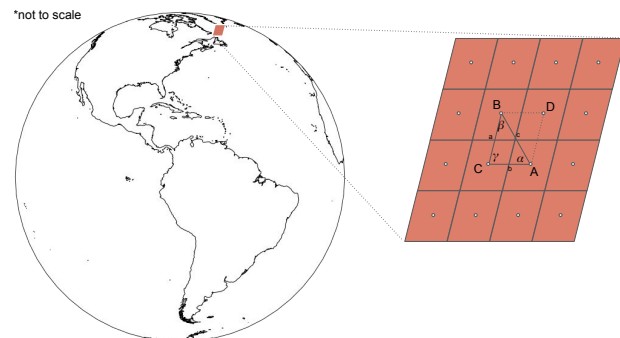

**Figure B1.** Notation used in formula for calculating GOES-16 ABI pixel area. The brown enlarged inset represents a portion of the ABI pixel grid with white dots on pixel centers.

## Appendix B: Pixel area calculation

GOES-16 ABI longwave pixels have a nominal (nadir) size of 2km per side but are larger than that at longer viewing angles. In this work we estimate the contrail longwave forcing in units of $GJ/km$ (Gigajoules per flight km) based on input quantities that are initially normalized by area (eg $W/m^2$) and so (both in synthetic validation tests to calculate ground truth and in the forcing estimates themselves) it is necessary to weight by pixel area. GOES-16 data are not provided with pixel area, but we utilize the latitude and longitude of pixel centers provided in the GOES-16 ABI L1b radiances data product to make a close

approximation of the pixel area for each pixel in the ABI pixel grid. Using the notation from Fig B1, we calculate the area of the pixel centered on $C$ based on the pixel centers $A, B, C$ making a triangle ($ABC$). The triangle side lengths $a, b$ are calculated using great circle distance along the surface of the earth as approximated by a sphere with radius=6371km. The interior angle $\gamma$ can then be found using the Law of Cosines (i.e., $\gamma = cos^{-1}(\frac{a^2+b^2-c^2}{2ab})$)) and finally the pixel area as a function of $a, b, \gamma$ comes from the formula for area of the parallelogram $ADBC = a \cdot b \cdot sin(\gamma)$.

*Code and data availability.* ERA5 data are available from the Copernicus Climate Change Service Climate Data Store (CDS): https://cds. climate.copernicus.eu/cdsapp#!/dataset/reanalysis-era5-pressure-levels?tab=overview. COIN data can be found at gs://upwelling_irradiance/ ceres_goes/. GOES-16 products can be found at: https://console.cloud.google.com/marketplace/product/noaa-public/goes. Collated dataset of sampled pixels can be found at gs://contrails_external/longwave_dataset/. Code with an example of a causal model regression performed on a collated dataset is available at https://github.com/google-research/google-research/tree/master/contrails_longwave/

*Author contributions.* **ASW**: Conceptualization, Software, Visualization, Writing - Review & Editing. **SG**: Conceptualization, Software, Writing - Review & Editing. **NG**: Software, Writing - Review & Editing, Supervision. **JN**: Software. **CVA**: Conceptualization, Writing - Review & Editing, Supervision. **KM**: Conceptualization, Software, Visualization, Writing - Original Draft, Writing - Review & Editing.



*Competing interests.* The authors declare the following financial interests/ personal relationships which may be considered as potential competing interests: Authors are employees of Google Inc. as noted in their author affiliations. Google is a technology company that sells

computing and machine learning services as part of its business.

*Acknowledgements.* The authors would like to gratefully acknowledge Tharun Sankar and Aaron Sarna for their software contributions that aided this work, John Platt for his insightful guidance and support, Sebastian Eastham for helpful early discussions, Dinesh Sanekommu for helpful comments on the manuscript, and Erica Brand and Rachel Soh for their contributions in data acquisition.





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
