# Peer review of "Observing long-lived longwave contrail forcing"

_EGUsphere, 2025_

## Referee Comment (RC1)

**General Comments**

This manuscript presents a causal-inference framework for estimating contrail longwave radiative forcing using GOES-16 OLR data, advected flight tracks, and regression against confounding variables. The approach is innovative, the manuscript is supported by appropriate references, and the synthetic dataset validation adds credibility.

At the same time, the framing of the new metric "oRF12" and several methodological details are not sufficiently clear. In particular, the analysis is restricted to longwave-only forcing, which risks being misinterpreted as a net effect. The role of the 12-hour window is not well explained upfront, and some key aspects of data handling and model comparison (cloud-phase confounding, CoCiP setup) are underspecified. These issues should be addressed before publication to ensure the results are reproducible and properly contextualized.

Overall, I find the manuscript to be a promising and potentially valuable contribution, but I recommend revision to sharpen the scope, clarify limitations, and provide more methodological detail.

**Specific Comments**

- 1. The manuscript refers to "long-lived contrail cirrus" (e.g., line 47) but does not provide an operational definition (e.g., > N hours). Longevity is only implied by the monotonic growth of oRFH up to 12 hours (Fig. 9). Please provide a clear definition and link it to how oRFH should be interpreted.
- 2. The paper emphasizes that the method does not require a contrail mask. In practice this means it is best applied to fleets or regional averages rather than individual flights. Please make this explicit and provide two or three concrete application scenarios (e.g., sectoral averages, model validation datasets).
- 4. The workflow (ADS-B + ERA5 + GOES-16 -> advection -> rasterization -> regression -> conversion to oRFH) is currently scattered across several sections. A simple flowchart would greatly improve clarity for readers and aid reproducibility.
- 5. The regression framework replaces the explicit "unaffected region" used in earlier studies. Please clarify more directly how the counterfactual is constructed statistically (i.e., which confounders are controlled, and how). It would also help to summarize the permutation test result in the main text, since this is crucial evidence against spurious correlation.

- 6. The paper uses GOES-16 COIN OLR as the outcome variable. Please consolidate the description into one place: which bands/channels are used, how COIN OLR is generated, and what assumptions might bias contrail-specific estimates.
- 7. The authors acknowledge that ERA5 alone cannot distinguish contrails from natural cirrus, and that GOES-16 cloud phase adds limited separation because contrails and natural cirrus both fall into the "ice" category. This is a central limitation. Please explicitly state in the manuscript that the method does not significantly improve contrail-cirrus separation, unless there is concrete evidence that it does.
- 8. Relatedly, Table 1 shows large coefficients in the clear-sky category, partly attributed to cloud-phase misclassification. It would strengthen the paper if the authors quantified the sensitivity of oRF12 to plausible misclassification rates.
- 9. The comparison between oRF12 and CoCiP longwave iRF is interesting but not fully contextualized. Please clarify which adjustment processes are captured by oRF12 within the 12-hour window and which are excluded relative to ERF.
- 10. The study focuses exclusively on longwave forcing. This should be emphasized more prominently in the Abstract, Introduction, and figure captions to avoid misinterpretation. At minimum, please provide an order-of-magnitude estimate or citation for shortwave effects in the study domain, so readers can understand whether oRF12 represents an upper bound of the net effect or only a partial contribution.
- 11. The CoCiP setup used for comparison is insufficiently described. Please state explicitly the interpolation method for meteorological inputs (linear, nearest, etc.), the model time step (10, 30, or 60 minutes), and any regional or temporal subsampling. These choices are critical for reproducibility and for interpreting differences between oRF12 and CoCiP estimates.
- 12. The manuscript does not provide any indication of computational cost. A simple case study (e.g., one day over CONUS) with approximate runtimes for advection, rasterization, and regression would help readers assess scalability and practical use.

**Technical Corrections**

- 1. Define acronyms ADS-B at first use.
- 2. In Fig. 10 (diurnal cycle), explain the longitude-to-local-time mapping and why the x axis spans 48 h.

- 3. What is the oRF3 in the caption of Figure 10 and Line 329? It is better to clarify again with H=3.
- 4. Abstract Line 4: "beyond a few hours" --> consider citing a specific range (3–6 h) with reference support.
- 5. Is it possible to provide more information in Section 3.3 for better understanding results in Fig. 9?

---

## Author Comment (AC1)

-Legend—Referee text is in blackAuthor response text is in blue

\_\_\_\_\_

**Referee #2**

**General Comments**

This manuscript presents a causal-inference framework for estimating contrail longwave radiative forcing using GOES-16 OLR data, advected flight tracks, and regression against confounding variables. The approach is innovative, the manuscript is supported by appropriate references, and the synthetic dataset validation adds credibility.

At the same time, the framing of the new metric "oRF12" and several methodological details are not sufficiently clear. In particular, the analysis is restricted to longwave-only forcing, which risks being misinterpreted as a net effect. The role of the 12-hour window is not well explained upfront, and some key aspects of data handling and model comparison (cloud-phase confounding, CoCiP setup) are underspecified. These issues should be addressed before publication to ensure the results are reproducible and properly contextualized.

Overall, I find the manuscript to be a promising and potentially valuable contribution, but I recommend revision to sharpen the scope, clarify limitations, and provide more methodological detail.

We thank the reviewer for their detailed review of the manuscript and helpful comments and suggestions. Please find below point by point responses.

**Specific Comments**

1. The manuscript refers to "long-lived contrail cirrus" (e.g., line 47) but does not provide an operational definition (e.g., > N hours). Longevity is only implied by the monotonic growth of oRFH up to 12 hours (Fig. 9). Please provide a clear definition and link it to how oRFH should be interpreted.

We have clarified our usage of 'long-lived' contrail cirrus is intended to mean both linear and non-linear contrail cirrus (in the abstract as well as first usage in the Introduction section).

2. The paper emphasizes that the method does not require a contrail mask. In practice this means it is best applied to fleets or regional averages rather than individual flights. Please make this explicit and provide two or three concrete application scenarios (e.g., sectoral averages, model validation datasets).

We have added this explicitly (along with some examples) to the "Limitations and Future Work" section.

<please note, there was no comment number 3 in the Reviewer comments provided to us by the
AMT review system>

4. The workflow (ADS-B + ERA5 + GOES-16 -> advection -> rasterization -> regression -> conversion to oRFH) is currently scattered across several sections. A simple flowchart would greatly improve clarity for readers and aid reproducibility.

Thank you for the suggestion, we have added a flowchart as figure 2.

5. The regression framework replaces the explicit "unaffected region" used in earlier studies. Please clarify more directly how the counterfactual is constructed statistically (i.e., which confounders are controlled, and how). It would also help to summarize the permutation test result in the main text, since this is crucial evidence against spurious correlation.

We have added more context and discussion in the Methods Section 2.1 on how Model 1 is constructing the statistical counterfactual for when there's no flight density (A=0), and for any counterfactual baseline flight density A=a, vs the observed A=a+d.

As the reviewer rightly notes, the permutation test is evidence against spurious correlation of our estimate. To make this more explicit, we have added a summary to the Insight from synthetic datasets section (Section 3.1), which discusses what particular coefficients are estimated to be zero in the model.

6. The paper uses GOES-16 COIN OLR as the outcome variable. Please consolidate the description into one place: which bands/channels are used, how COIN OLR is generated, and what assumptions might bias contrail-specific estimates.

We have consolidated the requested details about COIN OLR in section 2.2.1. Regarding the Referee's request for more details about how bias might affect contrail-specific estimates, we have augmented the "Limitations and Future Work" section to contain a general discussion of OLS regression input bias (including both COIN and confounder controls data).

7. The authors acknowledge that ERA5 alone cannot distinguish contrails from natural cirrus, and that GOES-16 cloud phase adds limited separation because contrails and natural cirrus both fall into the "ice" category. This is a central limitation. Please explicitly state in the manuscript that the method does not significantly improve contrail-cirrus separation, unless there is concrete evidence that it does.

We have added this clarification to the "Limitations and future work" section.

8. Relatedly, Table 1 shows large coefficients in the clear-sky category, partly attributed to cloud-phase misclassification. It would strengthen the paper if the authors quantified the sensitivity of oRF12 to plausible misclassification rates.

We agree that quantifying the sensitivity of our estimate to cloud phase misclassification strengthens the paper. To address this, we conducted a sensitivity analysis using our synthetic dataset where the ground truth forcing is known. We have added the results of this analysis as a new appendix in the revised manuscript (Appendix C: Sensitivity of oRFH=12 to Cloud Phase Misclassification), which includes a detailed explanation of the methodology and a new figure (Figure C1). We also refer to this analysis in the discussion of Table 1. The results show an inverse relationship between the accuracy of the cirrus cloud phase classification and the error in our forcing estimate. This provides quantitative support for our hypothesis that misclassification is a primary driver of the large coefficients observed in the clear-sky category in Table 1. The analysis also shows that our methodology is robust under realistic levels of misclassification.

9. The comparison between oRF12 and CoCiP longwave iRF is interesting but not fully contextualized. Please clarify which adjustment processes are captured by oRF12 within the 12-hour window and which are excluded relative to ERF.

We are not aware of any studies to date which have quantified the timeline of (individual or collective) adjustment processes of contrails. The most similar studies from which to speculate are possibly CO2 and sea surface temperature rapid adjustment studies such as <a href="https://journals.ametsoc.org/view/journals/clim/22/11/2009jcli2652.1.xml">https://journals.ametsoc.org/view/journals/clim/22/11/2009jcli2652.1.xml</a> which reports in some study variants that cloud rapid adjustment response was "statistically consistent with the equilibrium value" by day 5 after the perturbation. We have updated the manuscript to include further discussion of this.

10. The study focuses exclusively on longwave forcing. This should be emphasized more prominently in the Abstract, Introduction, and figure captions to avoid misinterpretation. At minimum, please provide an order-of-magnitude estimate or citation for shortwave effects in the study domain, so readers can understand whether oRF12 represents an upper bound of the net effect or only a partial contribution.

We have added emphasis throughout about this analysis being longwave only, and added a CoCiP shortwave forcing estimate for the same spatio-temporal region (and brief discussion of whether oRF12 represents an upper bound of the net effect) to the "Contextualizing the Forcing Estimate" section.

11. The CoCiP setup used for comparison is insufficiently described. Please state explicitly the interpolation method for meteorological inputs (linear, nearest, etc.), the model time step (10, 30, or 60 minutes), and any regional or temporal subsampling. These choices are critical for reproducibility and for interpreting differences between oRF12 and CoCiP estimates.

Thank you for catching this omission, we have added these details to the "Synthetic dataset validations" section.

12. The manuscript does not provide any indication of computational cost. A simple case study (e.g., one day over CONUS) with approximate runtimes for advection, rasterization, and regression would help readers assess scalability and practical use.

Thank you for the suggestion, we have added these details as a new subsection "Computational expense" in the Methods section.

**Technical Corrections**

1. Define acronyms ADS-B at first use.

**Done.**

2. In Fig. 10 (diurnal cycle), explain the longitude-to-local-time mapping and why the x axis spans 48 h.

**Done.**

3. What is the oRF3 in the caption of Figure 10 and Line 329? It is better to clarify again with H=3.

This is the observational RF for the cumulative first three hours of advection estimated using Model (3) setting H=3. We have changed the notation throughout to  $oRF_{H=3}$  instead of  $oRF_3$  to make this explicit.

- 4. Abstract Line 4: "beyond a few hours" --> consider citing a specific range (3–6 h) with reference support.
- 3-6 hours has been added as the range in the Abstract, supported by adding to the Introduction a sentence citing the e-folding time from (Vázquez-Navarro et al., 2015).
- 5. Is it possible to provide more information in Section 3.3 for better understanding results in Fig. 9?

We have expanded Section 3.3, please don't hesitate to ask if something is still unclear.

---

## Author Comment (AC2)

-Legend—Referee text is in blackAuthor response text is in blue

**Referee #1**

In this paper, a Sonabend-W et al. quantified the longwave radiative forcing of contrails at the top-of-atmosphere (TOA) based on GOES-16 satellite observations and ERA5 reanalysis, and estimated that the longwave radiative forcing of contrails is 46.9 GJ on average over the Americas. The authors apply causal inference to discern the effect of contrails while controlling for radiative and cloud confounders. The authors compared their results with CoCiP data, and illustrates how the longwave warming of contrails varies between day and night.

This method is plausible, but the results are unreasonable, so the authors should check the details to correct any potential mistakes. The paper might be accepted after addressing the following issues:

 According to Fig. 10, the new method yields a longwave forcing of zero at noon, which is not reasonable. Unless there is no contrail at all at noon (which is untrue), the longwave forcing of contrails should not be zero. Furthermore, the surface temperature is higher at noon-time, so theoretically the longwave forcing of a contrail should be significant. Therefore, the CoCiP RF is more reasonable than that calculated by the new method.

Note: The authors added some discussions to address this issue, but it is hard to believe a zero longwave contrail forcing at noon. The unrealistic zero forcing at noon might be induced by issues in the regression process (see the next comment).

We thank the reviewer for their insistence on this point. In developing our response to this, we have performed an audit on our code and data, investigated further using the synthetic test data, and have contemplated and discussed amongst the co-authors some additional recommended practices for applying causal inference techniques to subgroup analysis (here, the subgroups are the GOES-ABI grid pixel data broken into their local hours to form diurnal progression).

In so doing we have:

A) Uncovered a minor error in Table 1 coefficient values.

Prior to manuscript submission, we fixed an off-by-one coding error which had resulted in approximately a 1% error on the magnitudes of  $A_{H=11}$  values used as inputs to the causal regressions. At that time we regenerated all the manuscript figures, but unfortunately neglected to update the coefficients of Table 1 prior to submission. We have now updated the coefficients in this manuscript revision.

Please note, this was discovered in the course of our audit of the diurnal trends figure, but it did not in fact change the diurnal trend figure, because those causal regressions had already been regenerated with the correct  $A_{H=11}$  values before the initial submission. Nor did it change the central estimate of 46.9 GJ/km longwave forcing over the Americas, because the fix only changed the central estimate by 0.01 GJ/km.

B) Used the synthetic data to generate a "calibration curve" for the causal regressions, by systematically varying the magnitude of the ground truth effect size.

Doing this has uncovered an attenuation effect in the causal inference regressions: the smaller the magnitude of synthetic longwave forcing, the more severely the oRF estimate will under-estimate the magnitude. When the synthetic ground truth is approximately 10GJ GJ/km there is a noise floor, for which the oRF regression estimates are erroneously centered near 0 GJ/km. We have added a new Appendix D to the manuscript with the calibration curve figure and discussion.

We believe the physical phenomena or modeling artifacts we previously described in the manuscript could still be playing a role in driving  $oRF_{H=3}$  closer towards zero during midday hours than CoCiP's estimations. However, based on the confirmed magnitude of this effect in synthetic data, we now interpret this signal:noise ratio driven attenuation as the primary cause of the diurnal trends figure  $oRF_{H=3}$  estimates being close to zero in the midday local hours.

Note that our aggregated Average Treatment Effect central estimate of 46.9 GJ/km longwave forcing is situated in a relatively robustly calibrated portion of the calibration curve; we have added discussion of the the possibility that this estimate could be revised upward in magnitude with future work developing improved calibration of such causal regressions. However, we have intentionally *not* revised the central estimate based on this calibration curve in the abstract or elsewhere in the manuscript, because we would like to perform a more thorough exploration of causal model calibration methodologies, deferred to future work.

C) Determined that the diurnal trend figure is better placed in the appendix as an initial case study of causal inference subgroup analysis.

Splitting out the GOES-16 ABI pixel gridded data into groups based on the local hour of their observation constitutes a form of subgroup analysis, and we apologize that we have not been

able to perform sufficient analysis to confidently separate potential modeling artifacts (such as uncontrolled confounding) apart from potential physical phenomena (such as atmospheric dehydration) in this context.

In consideration of the scope of the subgroup analysis methodology we now wish to be able to apply, thoroughly explore and validate (details in the new Appendix E), we would like to respectfully request the consent of the Reviewers and Editor to largely defer this type of diurnal subgroup analysis to future work.

1. In Eq. (3), a simple linear regression is used to calculate the parameters in the equation. However, as the authors pointed out, the correlation between independent variables Ai and Aj is large, so linear regression is not valid in this case. If the authors keep using simple linear regression, then this equation should be rewritten.

We appreciate the reviewer's comment and agree that if multiple linear regression with collinear input variables were being used it would not be valid, but our use of single linear regression is valid in this case because we're using a single advected trace density variable per regression fit. To make this clearer we have defined a new variable  $D_h$  which represents the trace density that has been advecting for h hours (rounding down partial hours), where h is in (0, ..., 11). This lets us be more explicit about defining the cumulative average advected trace density as:

$$A_H = \frac{1}{H+1} \sum_{h \leq H} D_h$$
, for  $H = 0, \dots, 11$ .

This notation allows differentiating between advected trace density that has advected for approximately h hours vs. the cumulative variables used in our regressions. If we were to use the hourly advected trace density  $D_h$  rather than  $A_H$  in Model (3), it would instead be the following:

$$E[\mathsf{OLR}_{\mathsf{COIN}}|\mathsf{CP}, D_0, \dots, D_{11}, \mathsf{OLR}_{\mathsf{ERA5}}] = \alpha_0 \cdot \mathsf{OLR}_{\mathsf{ERA5}} + \sum_{j=0}^4 I_{\{\mathsf{CP} = j\}} \left(\beta_j + \sum_{h=0}^{11} \gamma_{jh} D_h\right).$$

However, this model is not valid because it suffers from the high degree of correlation between different  $D_h$  as noted by the reviewer. For this reason we suspect the reviewer's comment is referring to correlations between  $D_i$  and  $D_j$  rather than between  $A_i$  and  $A_j$  and we apologize for the lack of clarity in our notation here. To be very explicit, in our method we only ever fit a single regression at a time with advected trace density treatment  $A_H$  from one maximum advection age  $A_H$ , where  $A_H$  is a scalar value per pixel. For example, we are never fitting a linear regression on multiple correlated inputs  $A_{H=3}$  and  $A_{H=4}$ .

The approach we visualize in the violin plot figures with H on the x-axis (Fig 9 and 10 in the newly revised manuscript) is perhaps more similar to a sensitivity analysis or model comparison than to multiple linear regression; we are visualizing how the cumulative effect estimate evolves as the time window for advection expands. To this end, in the figures where these cumulative estimates are rendered, the Y-axis labels clearly describe them as a cumulative quantity, and we use violin plots to visually emphasize they are fitted with separate regressions. In this revision we have also augmented the Figure 10 legend to reiterate that it is a plot of a cumulative quantity and caution that the slope of the curve at any point may not be a good indicator of the marginal radiative forcing of contrails having that age.